# Lenalidomide plus Dexamethasone Combination as First-Line Oral Therapy of Multiple Myeloma Patients: A Unicentric Real-Life Study

**DOI:** 10.3390/cancers15164036

**Published:** 2023-08-09

**Authors:** Vittorio Del Fabro, Mary Ann Di Giorgio, Valerio Leotta, Andrea Duminuco, Claudia Bellofiore, Uros Markovic, Alessandra Romano, Anna Bulla, Angelo Curto Pelle, Federica Elia, Francesco Di Raimondo, Concetta Conticello

**Affiliations:** 1Division of Hematology with BMT, A.O.U. Policlinico “G.Rodolico-San Marco”, 95123 Catania, Italy; urosmarkovic09041989@gmail.com (U.M.); alessandra.romano@unict.it (A.R.); anna.bulla24@gmail.com (A.B.); angelocurtopelle@gmail.com (A.C.P.); fede.elia31@gmail.com (F.E.); diraimon@unict.it (F.D.R.); ettaconticello@gmail.com (C.C.); 2Division of Hematology, Azienda Ospedaliera di Rilievo Nazionale e di Alta Specializzazione Garibaldi, 95122 Catania, Italy; mdigiorgio@arnasgaribaldi.it (M.A.D.G.); veotta@arnasgaribaldi.it (V.L.); bellofioreclaudia@gmail.com (C.B.); 3Amyloidosis Research and Treatment Center, Foundation Istituto di Ricovero e Cura a Carattere Scientifico (IRCCS), Policlinico San Matteo, 27100 Pavia, Italy; 4Dipartimento di Chirurgia Generale e Specialità Medico Chirurgiche, University of Catania, 95123 Catania, Italy

**Keywords:** multiple myeloma, lenalidomide, dexamethasone, Len/Dex, first-line treatment, elderly and frail patients

## Abstract

**Simple Summary:**

For a few years, lenalidomide plus dexamethasone (Len/Dex) has become a new standard of care for newly diagnosed multiple myeloma (NDMM) patients who are not eligible for autologous stem cell transplantation. The FIRST trial showed that continuous therapy with Len/Dex is superior in progression-free survival (PFS) and overall survival (OS) compared to fixed treatment or triplet, leading to approval for NDMM. The knowledge of the safety and efficacy of Len/Dex in frail and ultra-frail patients is limited. This study evaluates the Len/Dex combination, correlating it with the prognostic impact of different variables on PFS and OS. Our real-world data report that an elderly and frail population, rarely included in randomized clinical trials, may benefit from Len/Dex combination, with an incidence of adverse events that is inferior to pivotal trials. Thus, the oral and self-administering outpatient Len/Dex scheme is a considerable treatment choice in patients with high ECOG or age >75 years old.

**Abstract:**

Based on the results obtained in clinical trials, the use of the combination of lenalidomide and dexamethasone (Len/Dex) has become a potential therapeutic choice for newly diagnosed multiple myeloma (NDMM) ineligible for autologous stem cell transplantation. This study evaluated 89 frail NDMM patients treated with first-line oral association. At the last follow-up, 34 out of 89 patients (38.2%) were alive, and 22 were still in treatment with Len/Dex. Among 73 evaluable patients who received at least two cycles, the overall response rate was 71% (N = 52). The disease control rate, defined as any level of clinical response to therapy, occurred in 71 patients (97%). We reported one or more adverse events of grade 3 or 4 (G3/4) in 65.2% (N = 58) of patients, with a prevalence of hematological toxicity (24 patients), leading to an overall discontinuation of treatment in two cases. In univariate analysis, high ISS, high serum β2-microglobulin, and creatinine clearance <30 mL/min negatively impact OS, while the depth of response positively impacts OS. Moreover, G3-4 anemia, ISS, frailty score, and ECOG negatively impacts PFS. In conclusion, elderly and more frail patients benefit from the Len/Dex combination also in the era of monoclonal antibodies, ensuring an increased PFS and OS in patients where the therapeutic choice is often limited and usually not very effective.

## 1. Introduction

Multiple myeloma (MM) is still considered an incurable disease. However, the introduction over the past 20 years of new drug classes, such as proteasome inhibitors (PIs) and immunomodulatory drugs (IMIDs), has dramatically improved patient survival [1]. With the increase in therapeutic options, the goal of stratifying patients by risk has become even more important to diversify therapeutic approaches according to the different characteristics of the disease, as well as of the patient.

Immunomodulatory drugs have expanded the therapeutic scenario of multiple myeloma since the early 2000s. The class progenitor was thalidomide, followed by second and third-generation drugs, respectively, lenalidomide and pomalidomide. These molecules exert multiple direct and indirect anti-myeloma effects, primarily through binding to cereblon (CRBN), part of an E3 ubiquitin ligase complex, and promoting the ubiquitination of the IKAROS and AIOLOS family transcription factors (IKZF1 and IKZF3) [2,3]. The clinical use of IMiDs in MM has significantly improved long-term survival and quality of life.

The therapeutic approach for a newly diagnosed multiple myeloma (NDMM) patient is guided by eligibility for transplant procedures to date.

In NDMM young patients (conventionally < 65–70 years), standard therapy involves induction therapy to reach a rapid and profound response, followed by high-dose chemotherapy and autologous stem cell transplantation (HDT-ASCT, single or tandem), and then by consolidation and maintenance therapy if feasible [4].

In the context of not-transplant-eligible (NTE) patients, several clinical studies have investigated the efficacy of different therapeutic schemes, based on a fixed term, like on VISTA [5] or SWOG-S0777 trials [6], or continuous therapy like on FIRST [7], establishing the best choice of care for NTE NDMM patients as bortezomib plus melphalan and prednisone (BorMP), lenalidomide plus dexamethasone (Len/Dex), and, in the USA, bortezomib plus lenalidomide and dexamethasone [8].

Subsequently, the ALCYONE trial randomized two groups of frail patients to receive standard treatment according to the BorMP scheme or BorMP associated with daratumumab (anti-CD38) [9,10]. This study demonstrated an increase in PFS in the D-BorMP arm of 71.6% versus 50.2%, with a median follow-up of 16.5 months. The overall response rate was 90.9% in the daratumumab group, compared to 73.9% in the control group (*p* < 0.001).

The role of daratumumab was also evaluated in the MAIA trial. A total of 738 patients with a median age of 73 years were randomized to receive continuous therapy with Len/Dex versus the Len/Dex association with daratumumab, confirming the role of the anti-CD38 antibody in ensuring a significantly longer PFS (median PFS D-Len/Dex not reached vs. Len/Dex 31.9 months) and more profound molecular response (negative MRD, 24.2% vs. 7.3%, *p* < 0.001), and the risk of disease progression or death was 44% lower in the D-Len/Dex group than in the Len/Dex group [11].

So, ALCYONE and MAIA trials effectively explored the daratumumab potentiality, and placed it in the armamentarium of NDMM transplant-ineligible patients, yet this was at the first line [12].

Attention should be assessed for the elderly and frailty patients, possibly identifying parameters and variables able to predict the efficacy and tolerability of different treatments. In fact, in this setting of patients, the role of the Len/Dex regimen could be a current valid therapeutic option. In a European real-world analysis, in which patients who received first-line lenalidomide-based treatment had a significantly longer median PFS compared to the bortezomib-based scheme (*p* = 0.002; median, 38.4 vs. 31 months, respectively), and a considerably longer time to second treatment (*p* = 0.006; median, 45.7 vs. 36.5 months, respectively) [13].

Despite this, knowledge about the safety and efficacy of Len/Dex in frail and ultra-frail patients in a real-life setting is lacking. Larocca et al. confirmed that dose-adjusted Len/Dex followed by lenalidomide maintenance without dexamethasone prolonged event-free survival and a superimposable PFS and OS in a setting of elderly intermediate-fit patients with NDMM [14]. Recently, Facon et al. also evaluated the role of ixazomib (a proteasome inhibitor) added to Len/Dex, not confirming further benefit in terms of PFS and OS [15].

Although the new first-line standard of care for NTE is daratumumab-based schemes, a collection of real-life Len/Dex-treated patients can be useful to explore all aspects of this scheme on a fragile population, to highlight its limits and potential, especially even in patients whose own choice, for poor clinical conditions or logistical reasons they prefer an oral therapeutic approach.

This study aims to evaluate the efficacy and tolerability of Len/Dex in a real-life population of fragile multiple myeloma patients outside of a controlled clinical trial.

## 2. Materials and Methods

### 2.1. Patients

From June 2016 to May 2022, 89 non-transplant-eligible NDMM patients received lenalidomide and dexamethasone at the Division of Hematology, A.O.U. Policlinico “G.Rodolico-San Marco”, of Catania, Italy, according to FIRST schedule. The choice to treat these patients with the Len/Dex schedule rather than with BorMP association was made based on the disease’s characteristics, the patient’s clinical conditions, and the patient’s ability to access the hospital. According to manufacturer’s guidelines, lenalidomide dosage was reduced in patients with low platelet count and/or renal failure. We evaluated the safety and efficacy of the Len/Dex combination and the prognostic significance of several parameters on PFS and OS according to the routine clinical practice of our institution. All patients had a measurable disease as defined by IMWG guidelines, and 73 of them (82%) received at least two cycles of lenalidomide and dexamethasone, and 46 patients (51.7%) at least six cycles.

All participants gave written informed consent, according to the Declaration of Helsinki. Lenalidomide was given in doses from 5 to 25 mg daily on days 1–21 of each 28-day cycle following manufacturer’s guidelines, and dexamethasone 40 mg weekly (for <75 years patients) or 20 mg weekly (for ≥75 years old patients) until progression. A total of 22 patients (24.7%) were treated with 5 mg of lenalidomide, 34 (38.2%) with 10 mg, 29 (32.6%) with 15 mg, and 4 (4.5%) with 25 mg. The dose was modulated during the follow-up according to the approved guidelines [16].

Concomitant medications included agents for thromboprophylaxis with low-dose aspirin for low-risk patients and with low-molecular-weight heparin or equivalent or oral anticoagulant therapy for high-risk patients and anti-infectious prophylaxis, consisting of trimethoprim and sulfamethoxazole 800 mg bis in die only two days per week and acyclovir 200 mg daily. Patients received subcutaneous filgrastim as part of a prophylaxis regimen when neutrophils count ≤1 × 10^9^/L.

In addition, patients enrolled in the study were candidates for annual vaccination against influenza and pneumococcal pneumonia. In the Len/Dex group, 31 patients (34.8%) received the yearly vaccination against flu and 21 (23.6%) against pneumococcal pneumonia. From March 2021 to February 2022, both doses of anti-SARS-CoV-2 mRNA vaccination were given to all patients, and 70% received the third dose.

Physical and laboratory examinations were conducted on day 1 of each cycle. In 48 (53.9%) patients evaluated before Len/Dex start, FISH analysis was available for the following alterations: gain 1q, del 1p, del 13q, t(4;14), t(11;14), t(14;16) and del 17p.

Cytogenetic high risk was defined as the presence of t(4;14), t(14;16), or del17p documented by FISH at any percentage level, according to International Myeloma Working Group (IMWG) criteria [17].

Patients were stratified into three frailty risk categories (fit, unfit, or frail) based on the results of the IMWG frailty score, combining the patient’s age with parameters derived by evaluation instruments of patient’s comorbidities (Charlson Comorbidity Index—CCI) and fitness (ADL—Activities of Daily Living; IADL—Instrumental Activities of Daily Living).

We additionally applied to our study population the simplified frailty score, elaborated by Facon et al. [18], which stratifies patients to age, CCI, and ECOG and divides them into two categories: frail and not frail.

Using these scores, 100% of patients in our cohort should be classified as frail.

### 2.2. Safety and Clinical Evaluation

Adverse events (AEs) were evaluated based on Common Terminology Criteria for Adverse Events v5.0 (CTCAE v5.0). Responses were assessed according to the International Myeloma Working Group (IMWG) response criteria. Clinical benefit rate (CBR) was considered the percentage of patients that had a response higher or equal to minimal response (≥MR); disease control rate (DCR) included those patients that had a response equal to or higher than stable disease (≥SD).

### 2.3. Statistical Analysis

Data were elaborated using SPSS Statistics version 26. Descriptive statistics were generated to analyze results, and a *p*-value < 0.05 was considered significant. Fisher’s exact test was used for nominal variables with two categories; the χ^2^ test was for nominal variables with more than 2 categories. The variables that resulted as being significant from univariate analysis were evaluated in multivariate analysis. Overall survival (OS) was calculated from the time of inclusion until the date of evaluation or death for any cause. Progression-free survival (PFS) was defined as the interval from the date of initiation of treatment with Len/Dex to the first day of progression; if no progression occurred, to the date of evaluation. The Kaplan–Meier test analyzed OS and PFS. Standard errors were calculated by the method of Greenwood; the 95% confidence intervals are computed as 1.96 times the standard error in each direction. For the multivariate analysis, the logistic regression method was used. The statistical significance level was set at the 95th percentile. Significance above the 99th percentile was highly significant. A linear regression model performed a correlation between PFS or OS and the best response cycle.

## 3. Results

### 3.1. Basal Features (Table 1)

More than half of the patients were males (N = 56, 62.9%), with a median age of 77 years old (range 41–92 years); 50 patients (56.2%) were at least 75 years old. All patients had a secreting MM, and 10 (11.2%) had micromolecular MM.

ECOG performance status was higher or equal to 2 in 67 patients (75%), while according to the IMWG frailty score, 6 patients (6.7%) were classified as fit, 28 (31.4%) as unfit, and 55 patients (61.8%) as frail (a score higher or equal to 2). Charlson’s comorbidity index was similar to or lower than 3 in 36 patients (40.4%), 33 (37%) had a score between 4 and 5, and 20 patients had a score equal to or higher than 6 (22.6%).

ISS stage at diagnosis was higher or equal to II in 68 (76.5%) patients at diagnosis; similarly, the R-ISS, although evaluable in 48 (53.9%) of patients, was higher or equal to II in 37 (41.6%) patients.

Focusing on 48 patients (53.9%) who had an evaluable FISH analysis, 11 of them (23%) were classified as high risk for the presence of del(17p) or t(4;14) or t(14;16) alone or in combination, while 29 (60%) carried acq1q.

Renal function was assessed using an estimated glomerular filtration rate (eGFR) that was normal in 43 (48.3%) patients and compromised in 46 (51.7%). Twelve patients (13.5%) had an eGFR below 30 mL/min.

**Table 1 cancers-15-04036-t001:** Basal clinical characteristics of 89 Len/Dex-treated patients.

	N = 89 Patients
- Age, median (range)	77 (41–92)
≤64 years, N (%)	4 (4.5%)
65–75 years, N (%)	35 (39%)
>75 years, N (%)	50 (56.2%)
- Gender, male/female, N (%)	56 (62.9%)/33 (37.1%)
- Previous smouldering MM	36 (40.5%)
- Paraprotein (isotype)	
κ-light chain/λ-light chain, N (%)	58 (65.2%)/31 (34.8%)
micromolecolar, N (%)	10 (11.2%)
IgA, N (%)	27 (30.3%)
IgG, N (%)	51 (57.3%)
IgD, N (%)	1 (1.1%)
- ECOG (Performance Status at baseline)	
0–1, N (%)	22 (25%)
2, N (%)	32 (36%)
3 or more, N (%)	35 (39%)
- Frailty score IMWG	
Fit (score 0)	6 (6.7%)
Unfit (score 1)	28 (31.4%)
Frail (score ≥ 2)	55 (61.8%)
- Charlson Comorbidity Index	
<3	36 (40.4%)
4–5	33 (37%)
≥6	20 (22.6%)
- Frailty score (Facon et al.)	
Non Frail (score 0–1)	0 (0%)
Frail (score ≥ 2)	89 (100%)
- ISS stage at baseline	
I, N (%)	21 (23.5%)
II, N (%)	28 (31.5%)
III, N (%)	40 (45%)
- R-ISS stage at baseline, if evaluable	
I, N (%)	9 (10.1%)
II, N (%)	24 (27%)
III, N (%)	13 (14.6%)
- Cytogenetics risk evaluable	48 (53.9%)
High, N (%)	14 (29.2%)
Standard, N (%)	34 (70.8%)
t(4;14), N (%)	3 (6.2%)
t(14;16), N (%)	4 (8.3%)
del(17p), N (%)	7 (14.6%)
Amp(1q), N (%)	28 (58.3%)
- Creatinine clearance	
<30 mL/min, N (%)	12 (13.5%)
30–50 mL/min, N (%)	31 (34.8%)
>50 mL/min, N (%)	46 (51.7%)
- Bone Lesions, N (%)	76 (85.4%)
- Extramedullary lesions	12 (13.5%)
- Bone Marrow Involvement ≥60%, N (%)	26 (29.2%)
- LDH increased, N (%)	28 (31.5%)
- β2-microglobulin increased (≥3.5 mg/L), N (%)	60 (67.4%)

Bone lesions, detected by conventional skeletal survey, Whole-Body Low-Dose CT (WBLD-CT), nuclear magnetic resonance (NMR) or PET (Positron Emission Tomography) analysis, showed more than three lesions in 50 patients (56.2%) before starting treatment.

A minority of patients had extramedullary disease (N = 12, 13.5%), located mainly in the para-osseous area in the form of locally diffused plasmacytomas.

Lastly, increased lactate dehydrogenase levels were detected in 28 patients (31.5%); higher serum β2-microglobulin levels were detected in 60 patients (67.4%).

At the time of the analysis, 73 patients (82%) have received at least two cycles.

### 3.2. Safety and Tolerability

The total number of patients reporting G3-4 adverse events (AE) was 58 (65.1%), for a count of 76 AE.

As shown in Table 2, the most frequent G3-4 hematological adverse events were: anemia (N = 21, 23.6%), neutropenia (N = 9, 10%), and thrombocytopenia (N = 3, 3.4%). Forty patients (45%) received granulocyte-colony stimulating factor when neutrophils count was lower than 1.5 × 10^9^/L, and fifty-nine patients (66%) received erythropoietin as supportive care.

Totally we reported 33 G3-4 hematological toxicities in 24 patients.

The most frequent G3-4 non-hematological adverse events included: infections (N = 15, 16.8%—above all in the first cycles and during active disease) including pneumonia (N = 3, 3.4%), fatigue (N = 13, 14.6%), thromboembolism (N = 5, 5.6%), skin alterations such diffuse erythema (N = 4, 4.5%), constipation (N = 2, 2.2%), diarrhea (N = 1, 1.1%), heart failure (N = 1, 1.1%) and acute renal failure (N = 2, 2.2%).

Other less severe adverse events included four deep vein thromboses, two neuropathies, and four glucose metabolism alterations. Three patients (3.4%) died of treatment-related causes: two cases were due to G5 lung infection after one and six cycles, respectively; one patient died of intestinal perforation after one cycle.

Three patients (3.4%) developed a second primary malignancy (squamous cell epithelioma, squamous cell carcinoma and bladder cancer after 17, 30 and 44 months, respectively), as widely described for immunomodulatory drugs [19]. In patients with serious adverse events, the dose of lenalidomide was reduced (26 patients, 23%) according to manufacturer’s guidelines, delayed (32 patients, 36%), or permanently interrupted (10, 11%). One patient temporarily interrupted lenalidomide and dexamethasone after the 30th cycle for nine months due to acute renal failure for unclear reasons, then resolved.

Table 2 describes tolerability, treatment exposure and adverse events.

**Table 2 cancers-15-04036-t002:** Tolerability, treatment exposure and adverse events for the 89 Len/Dex-treated patients.

- Tolerability	
Dose reduction, N (%)	26 (23%)
Dose delayed, N (%)	32 (36%)
Dose interruption, N (%)	10 (11%)
Deaths (no treatment-related), N (%)	52 (58.4%)
Deaths (treatment-related), N (%)	3 (3.4%)
- Hematological adverse events (grade 3–4), N (%)	33 (37%) over 24 patients
Anemia, N (%)	21 (23.6%)
Neutropenia, N (%)	9 (10%)
Thrombocytopenia, N (%)	3 (3.4%)
- Non-hematological adverse events (grade 3–4), N (%)	43 (48.3%) over 34 patients
Infection, N (%)	15 (16.8%)
Fatigue, N (%)	13 (14.6%)
Thromboembolism, N (%)	5 (5.6%)
Diffuse erythema, N (%)	4 (4.5%)
GI disorders, N (%)	3 (3.4%)
Acute renal failure, N (%)	2 (2.2%)
Heart failure, N (%)	1 (1.1%)
Secondary malignancies	3 (3.4%)

### 3.3. Efficacy

Overall, the median PFS was 14 months (range 7.5–20.4) (Figure 1A) and 20 months in 73 patients who completed at least two cycles of therapy (Figure 1B); 16 patients did not complete at least two cycles due to rapid disease progression or worsening clinical conditions.

Still, focusing on the evaluable patients, we obtained a significant difference (*p* = 0.05) in PFS analyzing the subgroups based on frailty score. The median PFS was 13 months (2.7–23.3) for frail patients, 47 months (10.9–83.1) for unfit and not was reached for fit patients (Figure 2).

At the time of the analysis, 34/89 (38.2%) were alive and still being followed at our institution, 22 (24.7%) were still on treatment with lenalidomide and dexamethasone, 10 (11.2%) progressed before and underwent a second-line treatment, while 2 (2.2%) discontinued treatment due to toxicity or other complications. Among 73 evaluable patients who received at least two cycles, the overall response rate (ORR, ≥PR) was 71%, considering 4 (5.5%) patients in Complete Response (CR), 23 (31.5%) in Very Good Partial Response (VGPR) and 25 (34.2%) in Partial Response (PR) (Table 3).

The Disease Control Rate (DCR, ≥SD) was high and occurred in 71 patients (97%), defined as any level of clinical response to therapy, including no further need for transfusions, stable disease or regression of pain. The clinical benefit rate (CBR), including patients obtaining equal or more than MR, was brought in 62 patients (84.9%). Considering clearance creatinine data, in our group of 8 patients with clearance < 30 mL/min and need for dialysis, we observed an mPFS of 4 months vs. 24 months of 65 patients with clearance ≥ 30 mL/min (*p* = 0.01).

We analyzed the course of the disease based on the deepness of response obtained by patients at any time. As expected, patients who obtained more profound responses had longer PFS (Figure 3), especially for those who obtained at least a PR.

Dividing patients based on ISS stratification at the onset of the disease, we observed a mPFS of 52 months for patients in stage I, 47 months (18.6–75.4) for patients in stage II, and 8 months (4.5–11.5) for stage III (*p* = 0.02).

Among the 22 patients still being treated, 21 obtained at least a PR starting from the first cycle with a median time to best response of three treatment cycles. One patient achieved a VGPR during his 34th treatment cycle. We also found a strong correlation between OS and the cycle at which the best response was obtained. Using linear regression analysis, for each month that the best response is increased, survival increases by 2.2 months, suggesting the utility of continuous therapy that allows deepened response over time.

Regarding lenalidomide, 28 patients reduced the doses due to different toxicities. No differences in terms of OS and PFS were found regarding the subanalysis for groups divided according to different baseline doses or modifies during the follow-up (*p* > 0.05).

### 3.4. Predictors of Response

Aiming to identify parameters that could affect disease outcome, univariate analysis of PFS was performed by evaluating patients’ age, sex, frailty score, ECOG, ISS, LDH, β2-microglobulin, cytogenetics, creatinine clearance, high-grade anemia, high-grade neutropenia, bone lesions, G-CSF and EPO use, the deepness of response, treatment delay and reduction. Regarding PFS, multivariate analysis showed that high serum β2-microglobulin (*p* = 0.002), creatinine clearance <30 mL/min (*p* = 0.04), G3-4 anemia in any phase of treatment (*p* = 0.05), ISS (*p* = 0.01), frailty score (*p* = 0.03) and ECOG (*p* = 0.01) negatively impact on PFS, while G-CSF (*p* = 0.002) use, response deepness (*p* = 0.000), therapy delay (*p* = 0.01) and use of erythropoietin positively impact on PFS.

No differences in response rate were due to the presence or not and site of extramedullary disease, hemoglobin level or lenalidomide dose reduction.

### 3.5. Overall Survival

At the last follow-up, 34 out of the 89 (38.2%) patients were alive and 55 (61.8%) died, among which 42 were in disease progression, and 10 of them had received second-line therapy, represented mainly by the combination of daratumumab and bortezomib, and dexamethasone. We recorded two deaths due to pneumonia and seven caused by other complications, mostly heart failure, and stroke. Eight patients were lost to follow-up and considered dead in the statistical analysis. Three patients discontinued treatment due to toxicities (neuropathy or cutaneous reactions), and four patients for PD: all decided to start palliative care.

The number of cycles administered ranged from a minimum of 1 to a maximum of 54 (median of 6 cycles). Among all alive patients, 22 (24.7%) were still in treatment with lenalidomide and dexamethasone and 12 interrupted treatments; respectively, 10 patients (11.2%) progressed and underwent a second-line treatment, and 2 patients (2.2%) discontinued the treatment due to toxicities or other therapy-related complications.

Considering our entire cohort, 50 patients (56.2%) reported biochemical or clinical progression during treatment. Among these patients, 10 were still followed at our institution, and 4 were lost to follow-up.

In univariate analysis, high ISS (*p* = 0.001), elevated serum β2-microglobulin (*p* = 0.004), creatinine clearance <30 mL/min (*p* = 0.000), grade 3–4 anemia at the onset of the disease (*p* = 0.009) and worse frailty score (*p* = 0.002) negatively impact on OS. In contrast, the deepness of response (*p* = 0.000) positively impacts OS in our series.

Median OS was 22 months (range 13.4–30.6) for all patients, while it was 27 months (range 21.2–31.7) for the 73 evaluable patients (Figure 4); in this evaluable group, median OS for fit patients and unfit patients was not reached, while for frail patients it was 22 months (11.8–32.2, *p* = 0.02) (Figure 5).

Of 73 evaluable patients, those with creatinine clearance <30 mL/min (8 patients) achieved 7 months of median OS (range 0–15.3) versus patients with creatinine clearance >30 mL/min who achieved 30 months of median OS (range 12–47.9) (*p* < 0.001). Also, ISS negatively impacts the OS (*p* = 0.001).

In addition, the appearance of G3-4 anemia negatively impacts OS (*p* = 0.002), while using G-CSF does not improve OS (*p* = 0.06). Even for OS, the deepness of the response had a significant positive impact (*p* < 0.001) (Figure 6), particularly for patients who achieved at least a partial response.

The latest data shows a strong correlation between OS and the cycle to which the best response was obtained. According to our analysis and based on a linear regression model, for each month increase in achieving the best response, survival increases by 2.2 months (Figure 7).

On the contrary, CCI, serum baseline albumin, hypercalcemia and LDH did not impact OS. Moreover, the cytogenetic risk in our series does not affect overall survival, probably due to the low number of evaluable patients.

## 4. Discussion

Since its approval following the FIRST trial results, Len/Dex combination therapy has represented a valid therapeutic alternative in a population of often particularly fragile patients such as not-transplant-eligible patients. Moreover, Len/Dex has become the therapeutic backbone to which other drugs can be added (e.g., bortezomib, carfilzomib, elotuzumab or daratumumab) to increase effectiveness.

Based on experimental data and clinical experience, in the last decade, the concept of continuous myeloma treatment is increasingly developing, as opposed to the original idea of fixed-duration treatment. This occurred in the context of transplant-eligible patients, with single-agent lenalidomide in post-ASCT maintenance and in the setting of NTE patients, both with continuous up-front therapy until disease progression and maintenance post-frontline therapy [20]. The central aims of long-term treatment are different: prolong disease control, improve PFS and OS, maintain a minimal disease burden, prevent end-organ damage and induce a more profound and more lasting remission with desirable MRD (minimal residual disease) negative status. At the same time, one of the fundamental requirements of continuous treatment is the ability to be tolerated for a prolonged period with little cumulative or chronic toxicity or adverse impact on a patient’s quality of life. For this reason, real-life studies are becoming increasingly important in clinical research, mainly because patients who are usually excluded from clinical trials due to their high-risk profile (age, comorbidities, fitness) and patients enrolled in clinical trials deserve to be treated.

Among the phase 3 studies, including NTE-NDMM patients, the FIRST trial enrolled the most significant number of patients (a total of 1623), stratified by age, renal function, disease stage and cytogenetic risk at onset, with frailty assessment based on EQ-5D questionnaire results, a standardized measure of health status designed by the EuroQol Group [21].

A subsequent retrospective analysis by Facon et al. stratified patients enrolled in the FIRST study into two risk categories (non-frail/frail) based on a frailty score that included age, Charlson Comorbidity Index and ECOG score, with a higher prevalence of the non-frail patient group (51% vs. 49%). Our cohort of patients is entirely frail according to this score.

Subgroup analysis with updated follow-up at 67 months showed that FIRST-included patients receiving longer continuous Len/Dex treatment are younger (<75 years), with ISS < 3, and standard cytogenetic risk. Regarding treatment efficacy, approximately 50% of patients treated with continuous Len/Dex achieved a response ≥VGPR, with better-quality responses observed in most patients receiving long-term treatment with Len/Dex ongoing and those at standard cytogenetic risk [22]. These results, therefore, demonstrate that better clinical characteristics and less aggressiveness of the disease at onset correspond to a better response to treatment with ameliorated clinical outcomes.

Compared to the pivotal studies, our statistics include a higher number of elderly patients (≥75 years, 56.2% vs. 35% in the FIRST trial), with ECOG score equal to or greater than 2 (75% vs. 22%), high frailty score (≥2, 61.8% vs. not evaluated), and worse renal function (creatinine clearance < 30 mL/min, 13.5% vs. 8%). In addition, about one-third of patients had a high bone marrow infiltration of MM plasma cells and started treatment with low hemoglobin levels.

Despite this high-risk profile for age and fitness, in our cohort of patients lenalidomide and dexamethasone combination was feasible and well tolerated, with a similar or even inferior hematological and non-hematological toxicity profile.

Regarding safety, it has been widely described that recurrent infections represent the leading cause of death in patients with MM, who have a 7 times higher overall risk of developing infections of a bacterial nature (e.g., pneumonia, septicemia) or a viral nature (e.g., Herpes Zoster, flu viruses, SARS-CoV-2) compared to the general population, despite the use of prophylaxis [23,24]. This is probably due to the suppression of standard cellular B- and T-function and to pharmacological treatments. In this real-life study, although a high percentage of bone marrow plasma cells infiltration, the incidence of G3-4 neutropenia was significantly lower than the incidence reported in the FIRST trial (10% vs. 28%). We can therefore suppose that G-CSF primary and secondary prophylaxis, as we have recently published [25], performed in almost half of the patients in our real-life cohort, can reduce the incidence of G3-4 neutropenia.

This hypothesis is supported by increased PFS in patients who use G-CSF in prophylaxis. Similarly, only 16.8% of patients experienced grade 3/4 infections, whereas in the FIRST trial they occurred in 29% of patients who received continuous Len/Dex. Probably both supportive care with growth factors, antibiotic, antiviral prophylaxis and vaccination against influenza and S. pneumoniae and rapid reduction of lenalidomide dose have a relevant role in decreasing the percentage of infections that mainly occur in the first 6 months of therapy, as well as in our series when the disease is active and needs to be managed for infectious complications [26]. Moreover, reducing infectious episodes, especially pneumonia (our cohort 3.4%, FIRST trial 8%) represents the leading cause of death and therapy discontinuation, has certainly protected this frail cohort of patients compared with the FIRST trial (Table 4).

In a similar manner to supportive care, in our cohort of patients, the delay in treatment (36%) for any cause positively affected the effectiveness of the treatment in terms of median PFS. It could therefore be considered as a form of recovery before resuming treatment. Moreover, treatment delay does not impact OS.

Vice versa, despite recovery with supportive care with erythropoietin growth factor, our cohort’s low hemoglobin level negatively impacts median OS (*p* = 0.009) and PFS (*p* = 0.01). These results with inferior toxicity rates are probably because within a protocol, at least two episodes of grade 3 toxicity are required before reducing the dosage, while in real life it can be reduced as early as the first episode, thus improving overall safety.

Regarding efficacy, the IMWG frailty score significantly impacts median OS. This is probably because, different from frail patients, who often receive only one treatment line in their life, fit and unfit patients, when clinical and/or biochemical response to Len/Dex was considered poor, promptly started a more intensive second-line therapy and were clinically more resistant to disease-related disorders. In several cases, the association daratumumab and bortezomib plus dexamethasone were used as second-line therapy.

In addition, in the two most represented IMWG categories [unfit (28, 31.5%) and frail (55, 61.8%)], treatment interruptions due to any G3-4 toxicity were significantly higher in frail patients than in unfit patients (15, 31.9% versus 4, 14.2% respectively).

At the time of the analysis, 10 unfit (35.7%) and 10 frail (18.2%) patients were still on treatment. A more significant number of treatment discontinuation burdens this latter group of patients due to disease progression.

These data are consistent with those obtained from the retrospective analysis by Facon, conducted however on patients enrolled in a clinical trial and therefore selected for age and comorbidity [18].

Moreover, as already confirmed in the univariate analysis (mOS 25 vs. 4, *p* < 0.001), impaired renal function (clearance creatinine <30 mL/min vs. ≥30 mL/min) was a negative prognostic factor for OS. This assessment was evidenced by Dimopoulos et al., which divided the FIRST patients on the basis of renal impairment. Compared with treatment with melphalan, prednisone and thalidomide, continuous Len/Dex confirmed a reduction in the risk of progression or death in the subgroups with absent, mild and moderate (HR = 0.67, 0.70 and 0.65, respectively), with the negative prognostic role for severe renal impairment (clearance creatinine <30 mL/min) [27].

The same evidence is reported comparing our cohort with the frailty subgroup analysis of MAIA. In this context, daratumumab associated with Len/Dex reached better results, with ORR superior to Len/Dex arm (*p* = 0.026), confirming the potential role of the association with antiCD38 [28].

These data confirm that Len/Dex association is a valid therapeutic scheme in real life, even in a population of patients classified as unfit or frail, with the possibility of deepening the response as the number of therapy cycles increases. In fact, among the 22 patients still being treated, 21 patients obtained a clinical response equal to or greater than a PR starting from the 1st cycle (with a VGPR obtained in one case even in the 34th cycle) with a median time to the best response of three cycles of treatment. We also found a strong correlation between OS and the cycle at which the best response was obtained. According to our analysis, for every month that the best response is increased, survival increases by 2.2 months, which confirms the usefulness of continuous therapy that allows deepening the response over time.

Nowadays, the subcutaneous administration of daratumumab as first-line therapy in combination with lenalidomide and/or bortezomib can facilitate the inclusion of a part of frail patients in a more effective treatment, given by the monoclonal antibody itself. However, many frail patients have serious difficulties accessing the hospital. Furthermore, we must consider that for most frail patients, it will be feasible to use only one line of treatment in their life, as demonstrated by several studies [29]. Based on the clinical characteristics of these patients or their reduced self-care skills, we can identify a subgroup of patients that can obtain a prolonged survival benefit from an oral outpatient self-administering treatment limiting hospital access as much as possible, an aspect particularly relevant and suggested by several guidelines in the SARS-CoV-2 pandemic era [30], evidencing that the demonstrated therapeutic efficacy is also a factor to be taken into consideration in frail patients where endovenous treatment is difficult to perform, being not only a therapy administered as palliation, but rather to seek a hematological response.

## 5. Conclusions

These real-world data report the use of the Len/Dex treatment in a population rarely included in randomized clinical trials. Treatment is manageable, and G3-4 adverse events are less than pivotal trials probably because of a wide use of supportive care such as growth factors, anti-infectious prophylaxis and vaccination. The progressive deepening of the response demonstrated a significant positive prognostic role, confirming the validity of continuous therapy in multiple myeloma, capable of obtaining and deepening long-term clinical responses. We conclude that ongoing Len/Dex treatment is valid for patients with high ECOG or advanced age who should no longer be managed with supportive care alone.

## Figures and Tables

**Figure 1 cancers-15-04036-f001:**
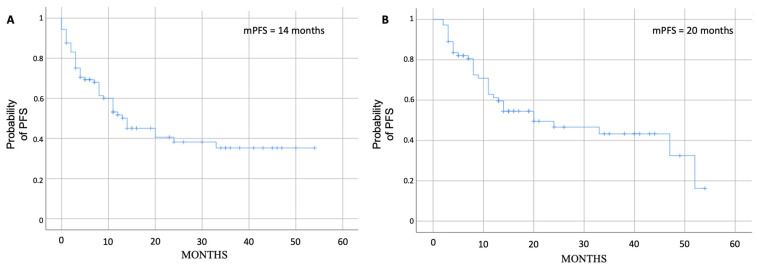
Median PFS for all 89 patients (**A**) and for the 73 evaluable patients with at least two cycles completed (**B**).

**Figure 2 cancers-15-04036-f002:**
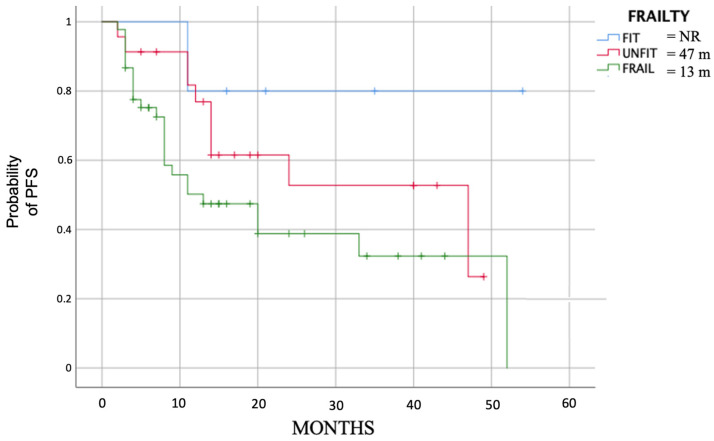
Median PFS (*p* = 0.05) divided by evaluable fit, unfit and frail patients.

**Figure 3 cancers-15-04036-f003:**
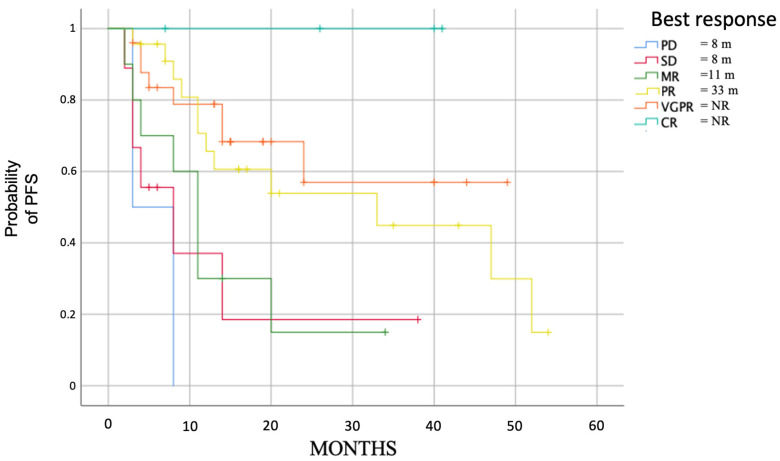
Median PFS in evaluable patients based on the deepness of response (*p* = 0.001).

**Figure 4 cancers-15-04036-f004:**
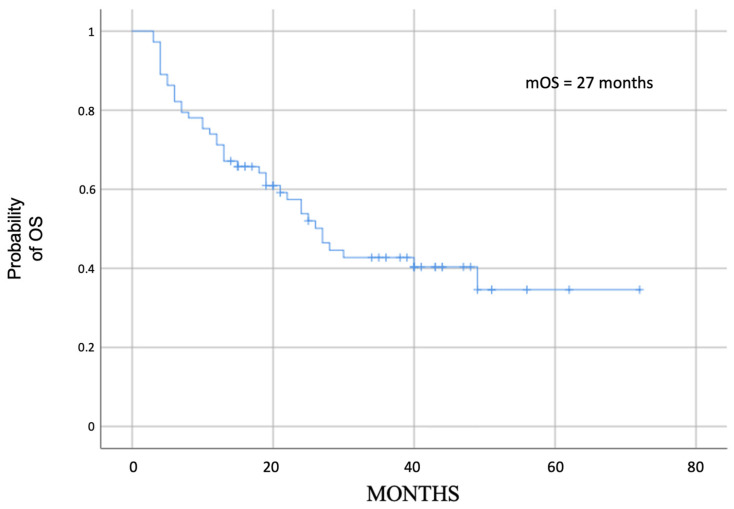
Median OS in all evaluable patients.

**Figure 5 cancers-15-04036-f005:**
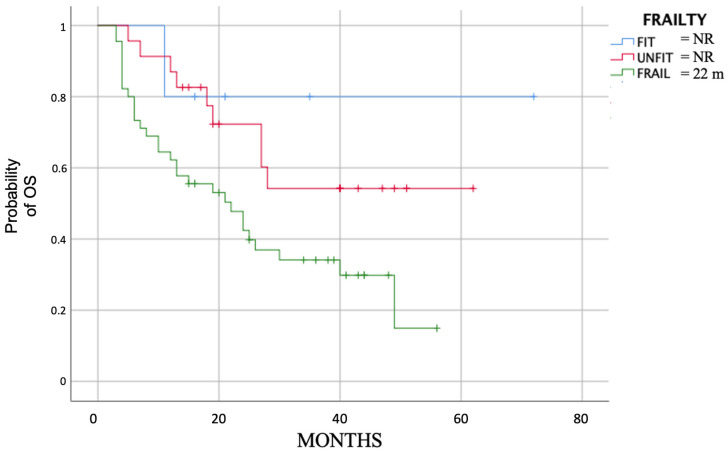
Median OS (*p* = 0.02) divided by evaluable fit, unfit and frail patients.

**Figure 6 cancers-15-04036-f006:**
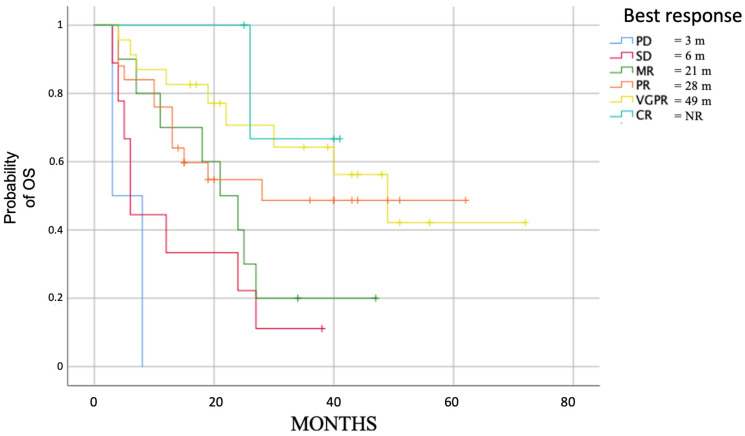
Median OS in evaluable patients based on the deepness of response (*p* < 0.001).

**Figure 7 cancers-15-04036-f007:**
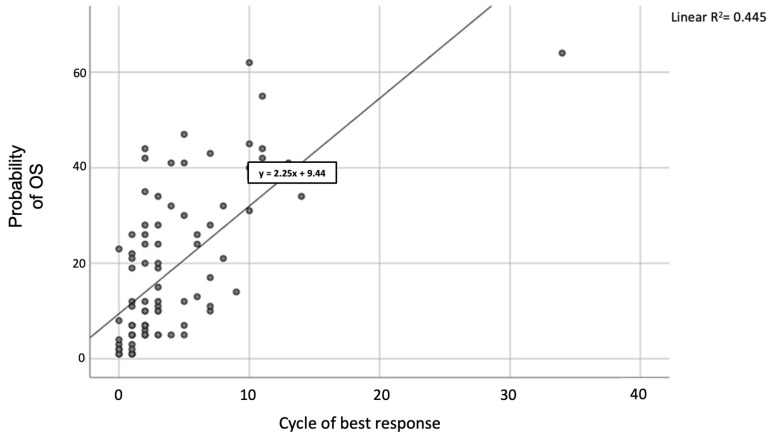
Linear regression between OS and the best response cycle in evaluable patients.

**Table 3 cancers-15-04036-t003:** Evaluation of efficacy. ORR (Overall response rate) ≥ PR; CBR (Clinical Benefit Rate); DCR (Disease Control Rate) ≥ SD.

	Best Response after Two or More Len/Dex Cycles N (%)	
CR	4 (5.5)	ORR 52/73 (71%) CBR 62/73 (84.9%) DCR 71/73 (97%)
VGPR	23 (31.5)	
PR	25 (34.2)	
MR	10 (13.7)	
SD	9 (12.4)	
PD	2 (2.7)	

**Table 4 cancers-15-04036-t004:** Comparison between Rd real-life data and Rd continuous arm of FIRST and frailty subgroup analysis of MAIA trials.

		Len/Dex Real-Life (73 Patients)	FIRST Trial Len/Dex Continuous Arm	MAIA Trial Len/Dex Frail Patients
Age (%)	≤75 years	43.8	65	27.2
	>75 years	56.2	35	62.8
Median PFS (months)	Fit	NR	Not evaluated	Not evaluated
	Unfit	47		
	Frail	13		
Median OS (months)	Fit	NR	Not evaluated	Not evaluated
	Unfit	NR		
	Frail	22		
ECOG ≥ 2 (%)		75	22	35
IMWG Frailty score ≥ 2 (%)		61.8	Not evaluated	Not evaluated
ISS III (%)		45	40	37
ClCr < 30 mL/min (%)		10.9	8	2
Neutropenia (grade 3–4) (%)		10	28	33
Infections (grade 3–4) (%)		16.8	29	28
Pneumonia (%)		3.4	8	10
Anemia (grade 3–4) (%)		23.6	18	25

## Data Availability

The data presented in this study are available on request from the corresponding author. The data are not publicly available due to privacy and ethical restrictions.

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
