# Peer review of "Lenalidomide plus Dexamethasone Combination as First-Line Oral Therapy of Multiple Myeloma Patients: A Unicentric Real-Life Study"

_cancers, 2023, doi:10.3390/cancers15164036_

Round 1
Reviewer 1 Report
Dear Dr. Fabro
I have reviewed the manuscript entitled “Len/Dex combination as first-line oral therapy of frail multiple myeloma patients: a unicentric real-life study”. In this manuscript, you analyzed newly diagnosed multiple myeloma (NDMM) frail patients treated with Len/Dex. You showed benefit from Len/Dex combination therapy and gave it as one choice of treatment in the era of monoclonal antibodies. The manuscript is well written, but the concept proposed by the authors is a little bit lacking in interest, and it is a low priority based on the novel and clinical nature of the finding report. Unfortunately, I think it is not suitable for this journal. Nevertheless, I described some comments below. If you are going to submit your paper to the other journal, you might like to revise your manuscript referring to my comments.
I have major comments.
1. You should describe the actual dose intensity of lenalidomide in the results and Table 1. How many patients could be given 25mg, 20mg, 15mg, 10mg, and 5mg of lenalidomide at first, respectively?
2. You compared your data only to the FIRST trial in the discussion. You should refer to the MAIA study's frailty subgroup analysis (Facon et al. Leukemia 2022,36:1066-1077). You can compare your data including efficacy, PFS, OS, and safety to the Rd Frail group in the MAIA study.
3. You mentioned “This assessment cannot be compared with the FIRST trial because the patient’s poor kidney function category is poorly represented” in lin432-433. However, there is a report about the impact of renal impairment (RI) outcomes with Len/Dex in the FIRST trial (Dimopoulos et al. Haematologica 2016,101(3):363-370). They demonstrated a PFS benefit could not be seen for Len/Dex continuous in the severe RI (CrCl,30 ml/min) group. You should add this report and discuss it in your discussion again.
4. In the subgroup analysis of MAIA, PFS in frail patients was not reached, and the ORR was 87% treated with Daratumumab plus lenalidomide and dexamethasone (D-Rd). Therefore, the clinical benefit of D-Rd was supported by NDMM patients regardless of frailty status. So, based on these discussions, you should consider the merit of Len/Dex not just the oral administration.
I have minor comments.
1. I think the colors of PR and VGPR are inverted in Figure 3.
2. You described “PFS” in the legend of Figure 6, but actual data showed “OS”? Which is right?
Reviewer 2 Report
The authors described real-world data of Len/Dex combination therapy for myeloma patients. This is a relatively small study, but real-world data is precious. I have coments.
1. In the title, "frail multiple myeloma "
The research includes fit , unfit patients at 6.7%, 31.4% by IMWG criteria , respectively.
2. In the title, Len/Dex → Lenalidomide and dexamethasone
is better.
3. In simple summary, the authors described almost all background. The authors should describe the summary of the research.
Reviewer 3 Report
This manuscript shows a retrospective analysis of frail patients with multiple myeloma (MM) who received Len/Dex therapy as a first-line regimen. There are few reports or studies for frail patients with MM, and this research will be important for better understanding of the practical therapeutic strategy for MM in the real-world. However, there are some concerns to be cleared in this manuscript as described below.
#1. I would like to confirm the definition of "PFS". In this manuscript, PFS was defined as "the interval from initiation of Len/Dex to the day of progression". How about the case of MM unrelated death, or initiation of second line therapies without progression? In the real-world setting, change of regimen without progression (due to adverse events or planed therapy) would occur frequently.
#2. There were no description for the initial dose of LEN. Were the initial dose decided by only renal function? Did all patients with no renal insufficiency receive full dose LEN? If any dose modification were underwent, discussions for adverse events should be modified.
#3. Labels and scales of Kaplan-Meier curve should be revised. Labels of Y-axis should be "probability of OS/PFS", and decimals (scales of X and Y axis) are unnecessary for Months.
#4. I fail to understand the meaning of figure 7 and associate analysis. I understand continuous treatment and deeper response would be associated with longer disease control. From the result of figure 7, the response from SD to MR at 6M would be superior than the response from VGPR to CR at 3M, is it right?
Reviewer 4 Report
The Article is well presented and statistically well processed.
Author Response
Thanks for your comment. We value your review
Reviewer 5 Report
This is a good, thorough and very detailed report of real world data in an elderly and frail population treated with lenalidomide and dexamethasone. Compared to trial populations this study is much more representative of the real world as it includes older, less fit patients and patient with significant renal impairment.
The paper is long and could be more concise and would benefit from that. It could also involve a discussion around ideal dose adjustment and whether genetic results should be used to inform treatment as these are the issues we need to address.
I do not think this paper adds much to the literature unfortunately as myeloma doctors are using this regimen in practice in this population, have treated 100-1000s of patients with it and already appreciate its efficacy and toxicity. There is sufficient data to suggest it is better tolerated than bortezomib based regimens. The biggest issue is around dose reductions and how best to do that. The MAIA study shows a huge benefit for patients who can tolerate len-dex-dara and is likely deliverable for most patients
Although they say treated as per FIRST trial it is unclear how strictly these patients were managed and if the exact parameters of that trial were followed for all patients. There is little discussion around genetics which was done in around half of the patients. Do the authors think genetic results should influence treatment in this population?
I am a bit confused about the claim that there is less toxicity in their study compared to the FIRST study as I cannot see a single p value and not sure how valid it would be to do this anyway between different trials – they need to be absolutely sure before making this claim. In addition to truly claim that they would need to show that patients were managed in completely identical way. The claim that their patients were better managed than the FIRST trial and hence had less infectious complications despite being older and less fit is very hard to prove and somewhat difficult to believe. These inter trial comparisons are very challenging but if the authors are going to do it then they need to look at MAIA trial and any other trial using lenalidomide-dexamethasone in a non transplant eligible population.
Round 2
Reviewer 1 Report
I have reviewed the revised version.
I understood the gist of this manuscript, and various issues were improved.
However, I feel short on interest, and it is unworthy of acceptance in this journal.
I leave the decision in charge of the editor.
Reviewer 5 Report
happy to accept